# A time-resolved multi-omic atlas of the developing mouse stomach

Xianju Li[1], Chunchao Zhang[2], Tongqing Gong[1], Xiaotian Ni[1,3], Jin'e Li[1], Dongdong Zhan[1,3], Mingwei Liu[1], Lei Song[1], Chen Ding[4], Jianming Xu[5], Bei Zhen[1], Yi Wang[1,2] & Jun Qin[1,2,4]

The mammalian stomach is structurally highly diverse and its organ functionality critically depends on a normal embryonic development. Although there have been several studies on the morphological changes during stomach development, a system-wide analysis of the underlying molecular changes is lacking. Here, we present a comprehensive, temporal proteome and transcriptome atlas of the mouse stomach at multiple developmental stages. Quantitative analysis of 12,108 gene products allows identifying three distinct phases based on changes in proteins and RNAs and the gain of stomach functions on a longitudinal time scale. The transcriptome indicates functionally important isoforms relevant to development and identifies several functionally unannotated novel splicing junction transcripts that we validate at the peptide level. Importantly, many proteins differentially expressed in stomach development are also significantly overexpressed in diffuse-type gastric cancer. Overall, our study provides a resource to understand stomach development and its connection to gastric cancer tumorigenesis.

[1] State Key Laboratory of Proteomics, Beijing Proteome Research Center, National Center for Protein Sciences (The PHOENIX Center, Beijing), Beijing 102206, China. [2] Alkek Center for Molecular Discovery, Verna and Marrs McLean Department of Biochemistry and Molecular Biology, Department of Molecular and Cellular Biology, Baylor College of Medicine, Houston, TX 77030, USA. [3] Department of Life Sciences, East China Normal University, Shanghai 200241, China. [4] State Key Laboratory of Genetic Engineering, Human Phenome Institute, Institutes of Biomedical Sciences, and School of Life Sciences, Zhongshan Hospital, Fudan University, Shanghai 200433, China. [5] Department of Gastrointestinal Oncology, Affiliated Hospital Cancer Center, Academy of Military Medical Sciences, Beijing 100071, China. These authors contributed equally: Xianju Li, Chunchao Zhang, Tongqing Gong  Correspondence and requests for materials should be addressed to Y.W. (email: yiw@bcm.edu) or to J.Q. (email: jqin@bcm.edu)

The mammalian stomach is a muscular sac[1] that has an evolutionarily diverse structure with versatile functions, including food digestion, hormonal regulation of metabolic homeostasis, and immune regulation[1–3]. The mouse stomach is composed of three main parts: the forestomach for food storage, the corpus for food digestion, and the antrum that secretes mucus and certain hormones[1,3–5]. The stomach originates from the embryonic foregut, and its ontogeny starts at approximately embryonic day 9.5 (E9.5)[6]. The stomach is protruding and visible at E10.5 and then rapidly grows[6,7]. Around E12.5, the stomach migrates to the left side of body; one day later, the forestomach starts to develop a stratified squamous epithelium, while the glandular stomach begins to develop a simple columnar epithelium[7,8] and divides into the corpus and antrum. Cells that originate from the corpus and antrum start to differentiate into smooth muscle, the muscle layer thickens at E14.5, and different types of cells ultimately become mature[1,6,9].

Currently, most studies of stomach development have focused on a single or a small set of proteins or on a specific pathway. Based on these efforts, the key roles of several transcriptional regulators (e.g., BARX[10], HOXA5[11], SOX2[1], CDX2[12], HNF1β[13], PDX[14], and GATA4[15]) and signaling pathways (Wnt[16,17], retinoic acid[18], Notch[19], FGF[1,3,4], BMP[20], SHH[15], Hippo[21], and TGF-β[22]) in controlling organ development have been defined. While our knowledge of stomach development is substantially enhanced by these studies, their limited focus hampers our understanding of the whole process in the context of systems biology, within which proteins are highly dynamic and interactive. Large protein sets can govern organ development by highly coordinated expression changes[23] and play important roles in organ functions.

Recently, significant advances have been made in mass spectrometry (MS)-based proteomics from sample preparation to liquid chromatography (LC) and instrumentation, making the identification of nearly all expressed proteins in cells or tissues possible with good accuracy and reproducibility[24–26]. In 2013, Geiger and his colleagues examined the proteomic profiles of 28 mouse tissues, including the stomach, using SILAC (stable isotope labeling by amino acids in cell culture) and LC-MS/MS (liquid chromatography tandem mass spectrometry) methods, which covered 7349 proteins[27]. Unlike studies on brain[28] and liver development[29], however, systematic mapping of the mouse stomach during development is missing at the proteome scale. A time-resolved, large-scale proteome atlas can answer unresolved and important questions, e.g.,: When does the expression of functionally important genes turn on? What expression trends do they exhibit? Who are the partners they function with? How do signaling pathways crosstalk? Are there distinct boundaries during development?

As mice are evolutionarily close to humans, and many well-known functionally important stomach proteins (e.g., PGC, ATP4A, and MUC5AC) are conserved in both species, it is a suitable model system for investigating organogenesis and human diseases, including cancer. For example, studies have confirmed that tumorigenesis shares many similar features with deregulated embryonic development[30,31].

In this work, we present a transcriptome-integrated proteomic analysis of the mouse stomach during organogenesis and development. We find similar characteristics in mouse stomach development and human gastric cancer tumorigenesis. Furthermore, our temporal proteomic and transcriptomic atlases identify three key phases during stomach development with distinct features. Moreover, integrative analysis of omics data sets reveals the dynamics of known and novel splicing events.

## Results

**A proteomic atlas of the developing mouse stomach.** To map the proteomic atlas of the developing mouse stomach, we collected three replicates of whole stomach organs at 15 timepoints that covered embryonic days (E12.5, embryonic day 12.5; E13.5, embryonic day 13.5; E14.5, embryonic day 14.5; E15.5, embryonic day 15.5; E16.5, embryonic day 16.5; E17.5, embryonic day 17.5; and E18.5, embryonic day 18.5), postnatal days (D1, day one; D3, day three; and D5, day five), and postnatal weeks (W1, week one; W2, week two; W3, week three; W6, week six; and W8, week eight). The sampling time window covered from the earliest day that stomach sampling is anatomically feasible to the latest weeks when stomachs are close to mature (Fig. 1a). We applied a fast-seq proteomic workflow[32] (Fig. 1b), a label-free quantitative proteomic approach in combination with a small-scale reversed-phase prefractionation strategy[33], and identified 12,108 gene products (GPs) at a 1% peptide FDR (false discovery rate) (Fig. 1c, Dataset1 in Supplementary Data 1). A total of 8,865 GPs were identified as high-quality IDs by selecting those that have been measured with at least one unique peptide (GPs-specific sequence) and two strict peptides (mascot ion score ≥ 20) (Dataset2 in Supplementary Data 1). Further filtering for proteins identified in at least 2 of the 3 replicates at one timepoint resulted in a final list of 7,678 GPs for bioinformatic analyses (Fig. 1c, d; Dataset3 in Supplementary Data 1). After these filtering steps, all experiments showed good consistency and a high degree of correlation ($R = 0.8$–$0.99$) between the temporally adjacent experiments (Supplementary Fig. 1a). The relative abundance (iFOT) of the proteins spans approximately eight orders of magnitude after logarithmic transform, which reflects the highly dynamic nature of the stomach proteome (Fig. 1e). In general, housekeeping proteins (e.g., TUBB4B and HSP90B1) show stable expression across the whole developing stages, whereas proteins related to liver development (e.g., AFP and ALB) are also highly dynamic and time-dependent, which suggests their widespread roles in organ development[34] (Fig. 1f). A list of 2890 proteins (Supplementary Fig. 1b) is compiled from all 15 timepoints that may represent the core components of the stomach proteome. The functions of these proteins are significantly enriched in multiple pathways, including metabolism, spliceosome, protein processing, and other pathways (Supplementary Fig. 1c).

**Three distinct phases of the developing mouse stomach.** The development of the stomach is tightly regulated by a series of signaling events and clusters of effectors on key pathways to control endoderm patterning, gastric specification, stomach regionalization, and morphogenesis[35]. The changes in protein machinery determine the fate of organogenesis and the development of the stomach. Although the whole developmental process changes gradually, boundaries (or phases) that separate critical events may exist that indicate profound changes during development. To investigate these events at the protein expression level, we carried out two independent statistical analyses, namely, an unsupervised hierarchical clustering analysis (HC) and principal component analysis (PCA), to identify discernable boundaries that demarcate the key stages with distinct features. Both algorithms reached a consensus on partitioning the experiments into three distinct phases: Ph1 (E12.5–E16.5), Ph2 (E17.5–W2), and Ph3 (W3–W8) (Fig. 2a). To identify representative proteins in each phase and understand their biological significance, we performed protein coexpression analysis and KEGG (Kyoto Encyclopedia of Genes and Genomes) pathway analysis on 4,300 differentially expressed proteins in the three phases (ANOVA, FDR < 0.01, Supplementary Data 2) (Fig. 2b, c). The KEGG analysis suggested that Ph1 is associated with cell division. Ph1

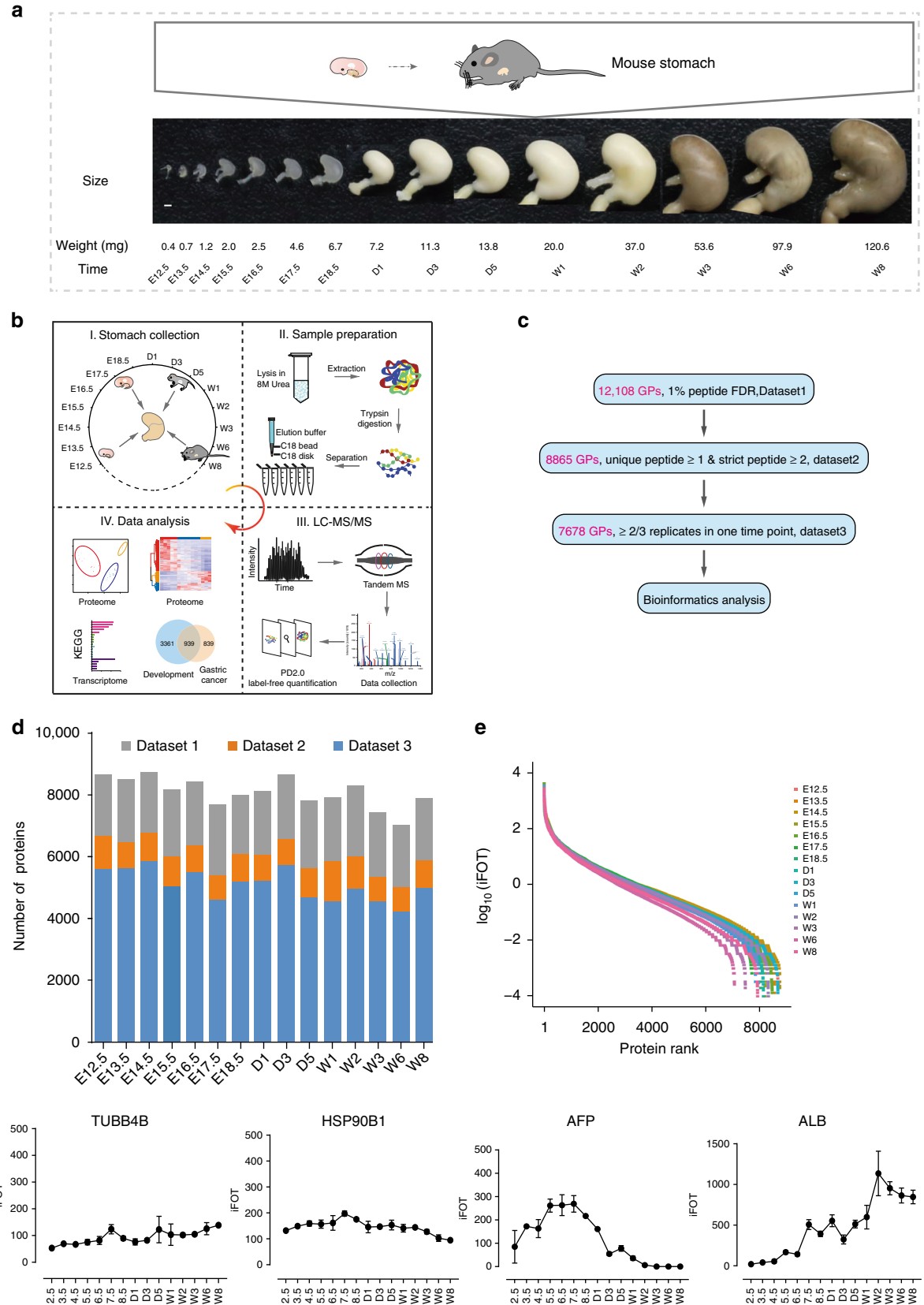

**Fig. 1** Proteomic landscape of mouse developing stomachs. **a** Illustration of mouse stomachs during development and 15 sampling timepoints. **b** The general workflow of MS-based quantitative proteomic and bioinformatic analyses. **c** Multiple data sets with different filtering criteria. **d** The total number of gene products identified in each timepoint. **e** Dynamic ranges of the mouse stomach proteomes measured at 15 timepoints. **f** The expression profiles for four proteins at 15 timepoints. Error bars are created by mean ± SD (standard deviations) of three replicates. Scale bar in **a** represents a unit of 1 mm

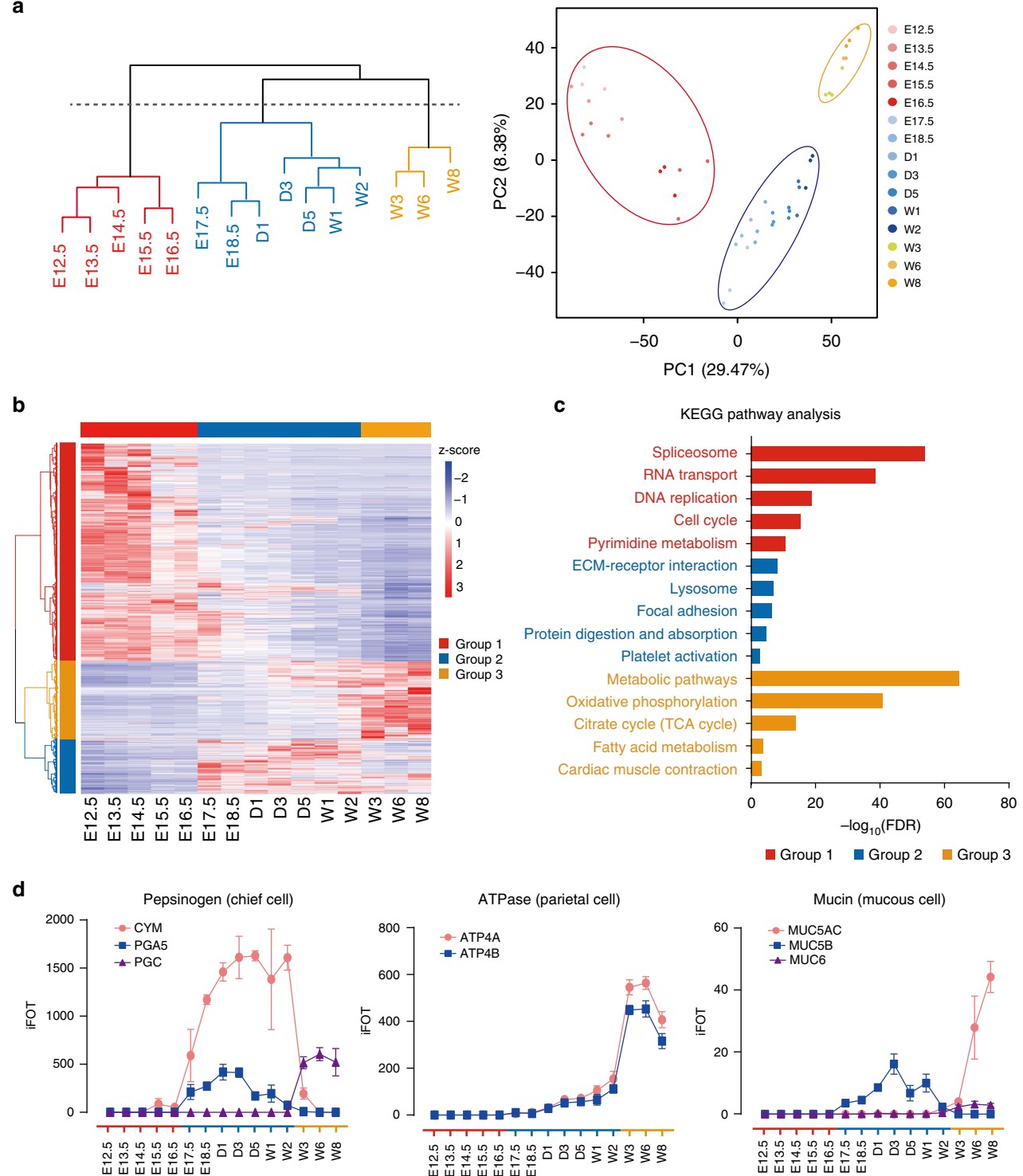

**Fig. 2** Three developmental phases and their biological functions and features. **a** Hierarchical clustering and PCA analyses of temporal proteomic data to separate the whole developmental process into three phases. **b** Coexpression analysis of 4300 differentially expressed proteins by semi-supervised, Ward's hierarchical clustering. **c** KEGG pathway analysis of three protein groups. **d** Changes of three protein families and eight individual stomach markers across 15 timepoints or three developmental phases. Error bars are created by mean ± SD of three replicates. **a** red, Ph1; blue, Ph2; and orange, Ph3. **b** horizontal color bars: red, Ph1; blue, Ph2; and orange, Ph3. vertical color bars: red, group 1; blue, group 2; orange, group 3. **c** color bars: red, pathways enriched in group 1; blue, pathways enriched in group 2; and orange, pathways enriched in group 3

has the greatest number of differentially expressed proteins, which are mainly enriched in pathways including spliceosome, RNA transport, DNA replication, cell cycle, and pyrimidine metabolism (Fig. 2c). Ph2 covers the perinatal and lactation period and is the phase at which the stomach increases most in

shape and mass. Ph2 has the least number of highly expressed proteins (proteins that are more highly expressed in one phase than in the other phases) and is enriched in ECM-receptor interaction, lysosome, and focal adhesion (Fig. 2c). Ph3 is the mature phase as the stomach has gained metabolic functions

(Fig. 2c). The boundary between Ph2 and Ph3 lies in W3 when infant mice are weaned (20–22 days after birth) and begin to eat food[36].

**Dynamics of key gastric proteins during stomach development**. As a major digestive organ, the stomach produces many digestive enzymes to breakdown food into small molecules for easy absorption. We subsequently surveyed three important protein families and traced their dynamic changes during the development. Pepsinogens, the precursors of pepsins, are digestive enzymes synthesized and secreted by gastric chief cells[37]. Pepsinogens A (PGA), B (PGB), F (PGF), progastricsin (PGC), and prochymosin (CYM) are five types of zymogens of pepsins almost exclusively expressed in the stomach of the tetrapod[38–41]. Our proteomic data identified three of them (CYM, PGA5, and PGC) with different trends of expression (Fig. 2d). CYM and PGA5 were highly expressed in Ph2, while PGC was only detected in Ph3. As mice in Ph2 were feeding on milk, the high levels of CYM and PGA5 indicate that they may play important roles in dairy protein digestion. Moreover, PGC was not detectable until W3 and was maintained at high levels after W3. This finding agrees with the expectation that PGC is predominantly expressed in the adult stomach[38].

Pepsinogens are activated in an acidic environment created by the gastric proton pump, an $H^+/K^+$-ATPase present in the parietal cells of the gastric oxyntic mucosa. The $H^+/K^+$-ATPase has two subunits (ATP4A and ATP4B), and it is one of the critical components of the ion transport system that mediates acid secretion[41–43]. The expression levels of both ATP4A and ATP4B were concurrently increased in Ph2 and peaked in Ph3 (Fig. 2d), marking the formation and maturation of the parietal cells in Ph2 and Ph3 as the mouse stomach matures.

To survive in the acidic environment, the stomach secretes mucin proteins to protect itself from damage. These proteins are the main components of the gastric mucus layer and protect the underlying epithelium from biochemical and mechanical aggressions[44,45]. Of all 21 mucins, five mucins (MUC1, MUC4, MUC5AC, MUC5B and MUC6) were identified in this study (Fig. 2d, Dataset2 in Supplementary Data 1). MUC1 and MUC4 were expressed at very low levels compared with the other mucins and were only detectable in a few samples; MUC5B was mainly identified in Ph2, whereas the known gastric marker mucins[46] MUC5AC and MUC6 that were secreted by surface mucous cells and mucous neck cells were highly expressed in Ph3. These results suggest that the mature stomach produces mucins to protect against acidic and mechanical damages.

There are other proteins highly expressed in the stomach (ANXA10, CLDN18, CTSE, GHRL, GIF, TFF1, and VSIG1) that maintain normal stomach functions (Supplementary Fig. 2). Among them, GIF is a glycoprotein produced by the chief cells of the stomach. ANXA10 is a calcium- and phospholipid-binding protein expressed in the gastric mucosa[47]. VSIG1 is a cell adhesion protein required for the proper differentiation of glandular gastric epithelia[48]. CLDN18 is a gastric epithelium-associated claudin[49]. The gradual increase in these proteins in the Ph3 of development suggests that they may be markers of a mature stomach[50].

**Identification of differentially expressed splicing isoforms**. In parallel with MS-based proteomics, we also carried out high-throughput RNA-Seq on mouse stomach tissues collected at the same timepoints. Our data analysis pipeline enabled the identification of 16,839 genes with maximal FPKM (Fragments Per Kilobase Million) values ≥ 1 (Supplementary Fig. 3a and Supplementary Data 3). In general, RNA-Seq identified more genes than

did MS (Supplementary Fig. 3b), and a total of 10,994 genes were measured in both data sets (Supplementary Fig. 3c).

Hierarchical clustering analysis of the transcriptomic data separated the whole developmental process into two groups (Fig. 3a, left panel). The first big branch of the dendrogram may be further cut into two trees to separate the embryonic and postnatal days, by which all 15 timepoints may be grouped into three phases. The same classification was also supported by the PCA analysis (Fig. 3a, right panel). This classification overlapped almost completely with the 3 phases obtained from the proteomic data with only one timepoint (E17.5) difference from the transcriptomic based classification. These observations indicate that the developmental process from E12.5 to W8 may be roughly divided into three phases based on either protein or RNA expression levels.

Semi-supervised hierarchical clustering analysis assigned genes into four groups based on their temporal expression patterns (Fig. 3b). The majority of the genes that were differentially expressed in the first phase were enriched in several pathways, such as spliceosome, RNA transport, and cell cycle (Fig. 3c). The second phase contains two gene groups involved in focal adhesion, ECM-receptor interaction, several signaling pathways, and fatty acid or amino acid degradation. The third phase is mainly associated with oxidative phosphorylation and metabolic activities. Overall, the RNA-Seq-based clustering analysis revealed three phases with similar biological features to those obtained from the proteomic data.

Previous studies have revealed the roles of splicing isoforms in various biological processes, including organ development[51]. The RNA-Seq data also enabled the assessment of splicing isoforms with deep coverage. When filtering by FPKM ≥ 1, 24,346 isoforms from 7769 genes were called, with 2,989 isoforms (~12%) differentially expressed across the three phases (RNA-Seq-based clustering) (Supplementary Fig. 3d and Supplementary Data 4). Among these well-known pathway regulators or transcription factors, not all variants significantly changed, and some of them had no (or very low) expression at the mRNA level (Fig. 3d). These observations suggest that only a fraction of splicing isoforms are expressed at the protein levels and are functionally involved in stomach development.

**Protein–RNA correlation and novel splice-junction events**. Recently, genomic and proteomic studies have suggested that the levels of protein and mRNA are not well-correlated despite the fact that proteins are translated from mRNA[52]. For this reason, protein levels cannot be reliably predicted by gene expression. Consistent with previous findings, the correlation of mRNA and protein in this study was also moderate (Median Pearson correlation coefficients = 0.55) (Fig. 4a). Genes with positive correlations were mainly enriched in pathways including spliceosome, valine, leucine and isoleucine degradation, DNA replication, and proteasome (Fig. 4a), while genes with poor or even negative correlations were enriched in amino sugar and nucleotide sugar metabolism (Fig. 4a). As previously discussed, a substantial number of genes or proteins were only identified in one of the data sets (Supplementary Fig. 3c). The discrepancies between mRNA and protein may be attributed to complicated post-transcriptional regulatory mechanisms, different RNA or protein degradation kinetics, and the detection limits or biases of instruments in measuring mRNA and protein, among other reasons.

The good coverages of the transcriptome and proteome provide an opportunity to observe and compare the dynamic changes of important protein machinery at the RNA or protein levels. For example, the RNA-Seq results revealed that the

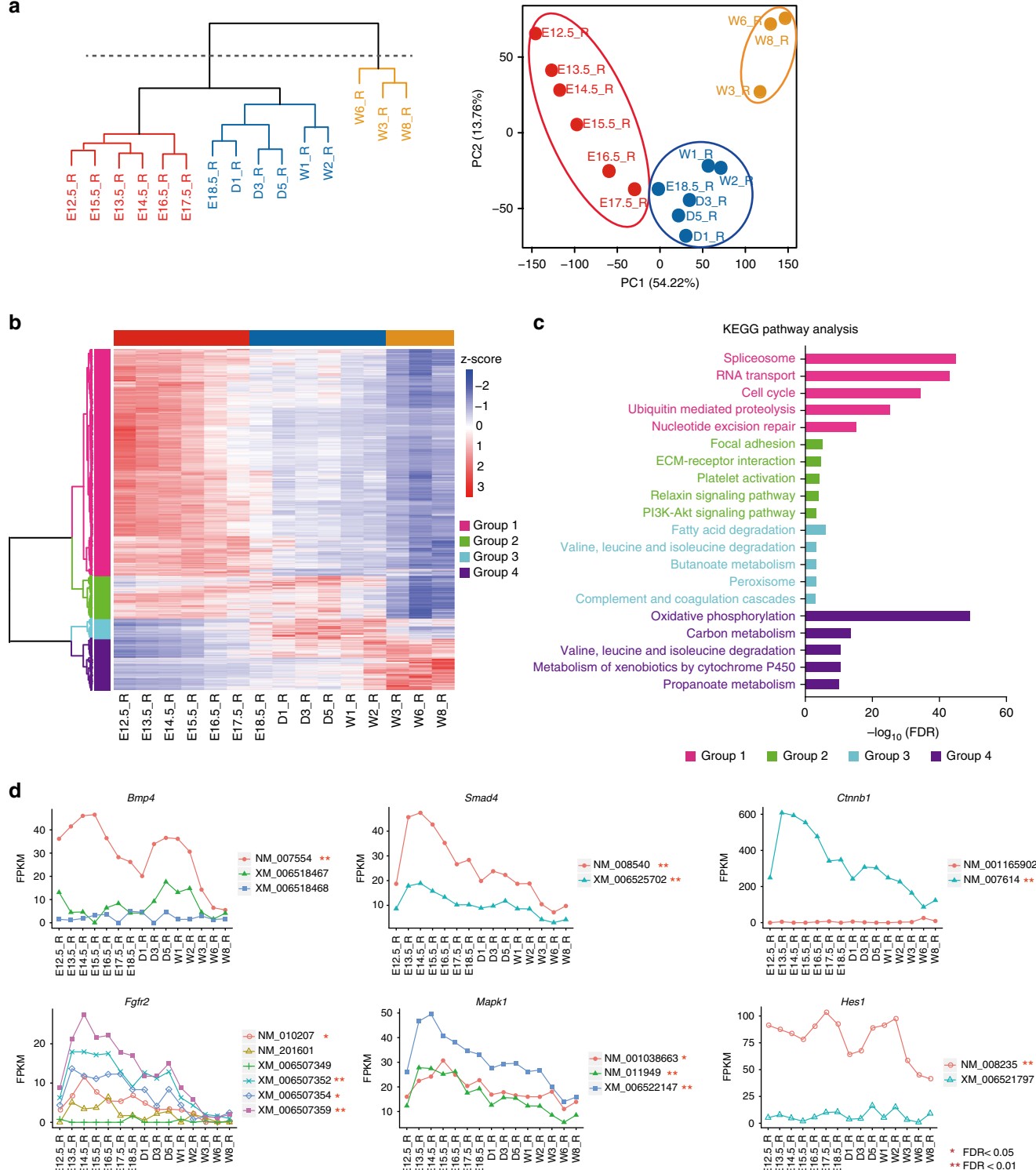

**Fig. 3** Transcriptomic landscape of developing mouse stomachs. **a** Hierarchical clustering and PCA analyses of temporal transcriptomic data. **b** Coexpression analysis of 11,044 differentially expressed genes by semi-supervised, Ward's hierarchical clustering. **c** KEGG pathway analysis of four gene groups measured by RNA-Seq. **d** Temporal transcriptomic profiles of six genes and their isoforms. **a** red, Ph1; blue, Ph2; and orange, Ph3. **b** horizontal color bars: red, Ph1; blue, Ph2; and orange, Ph3. vertical color bars: pink, group 1; green, group 2; cyan, group 3; and purple, group 4. **c** color bars: pink, pathways enriched in group 1; green, pathways enriched in group 2; cyan, pathways enriched in group 3; and purple, pathways enriched in group 4

expression patterns of the three protein families (pepsinogens, ATPases, and mucins) are similar to those measured by MS (Supplementary Fig. 4a). The proteasome is a multisubunit protein degradation machinery that plays diverse roles in the control of many basic cellular activities, including development[53].

Of the 41 major components (including 20s, 19s, 11s, and PA200), most of these components were highly expressed at early stages (Ph1), with the exception of a well-separated cluster that was differentially expressed in Ph2 and Ph3 in both the proteome and transcriptome (Supplementary Fig. 4b). This cluster contains

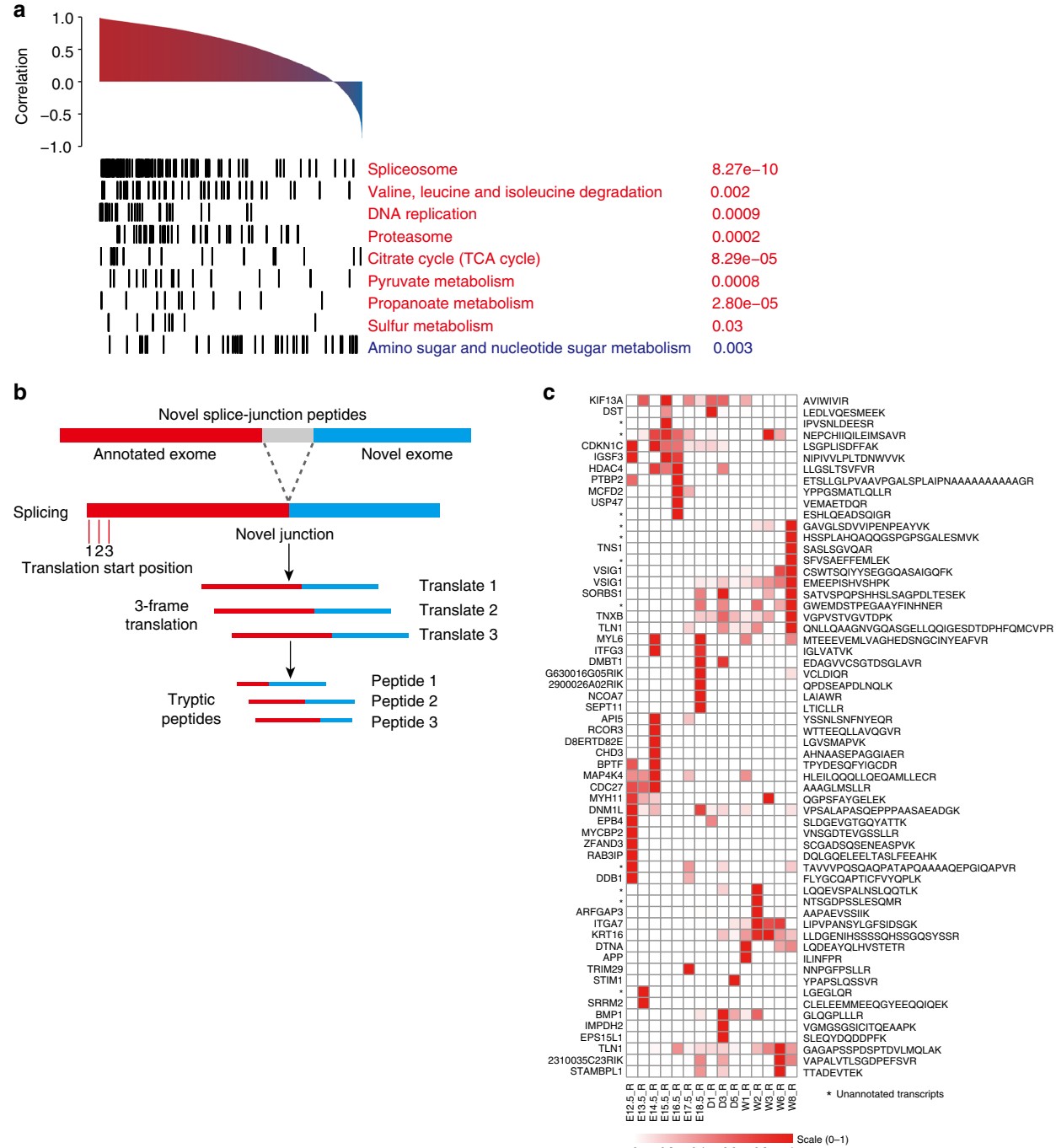

**Fig. 4** Integrative analyses of the mouse stomach proteome and transcriptome. **a** Protein–RNA correlation was calculated via Pearson's correlation coefficients using normalized scores (z-scores) across each gene product in all timepoints. Positively correlated genes and their enriched pathways are shown in red, and negatively correlated genes and their enriched pathways are shown in blue. **b** A pipeline to discover novel splice-junction peptides. **c** The relative abundances of 60 novel splice-junction peptides across all 15 timepoints. Stars are unannotated transcripts

three alternative β-subunits (PSMB8, PSMB9, and PSMB10) and two 11s subunits (PSME1 and PSME2). The incorporation of the alternative β-subunits PSMB8, 9, and 10 (also referred to as b5i, 1i, and 2i, respectively) in the 20s core and its regulatory particle 11s constitute a modified proteasome termed the immunoproteasome, which plays an essential function of processing class I major histocompatibility complex (MHC) peptides. The elevated expression of the immunoproteasome at Ph2 and Ph3 suggests its requirement in the maturation of the immune system. Interestingly, the γ-subunit of 11s (PMSE3, also referred to as REGγ) is

separated from this cluster and exhibits its highest expression at E12.5 in both the proteome and transcriptome. The 11s particle is a heptamer composed of seven molecules of the proteasomal subunit PSME3 or heterodimers of the subunits PSME1 and PMSE2. The distinct expression pattern of PSME3/REGγ implicates its critical role in early stomach development. Moreover, overexpression of REGγ has been reported in various cancers[54,55].

The Mediator complex is a transcriptional coactivator that plays a regulatory role in cell lineage development by modulating

transcription[56]. A total of 31 components from four submodules (head, middle, tail, and kinase) (Supplementary Fig. 4c) were observed with the highest expression recorded in Ph1, consistent with their role in RNA-associated transcription and cell division in the early developmental phase (Supplementary Fig. 4c). Two subunits of the middle module, MED14 and MED21, were also shown to express at high levels in late Ph1 or early Ph2, respectively. A previously published in vitro reconstitution experiment identified MED14 as both an architectural and a functional backbone of the Mediator complex[57], and its incorporation into the complex substantially enhanced the association of the mediator complex with RNA polymerase. MED21 is located at the boundary between the middle and head modules, suggesting that it may have a structural role in bridging the two modules in the Mediator complex. The increased expression of these two subunits indicates a possible functional requirement for the transcription of genes critical for the transition from Ph1 to Ph2.

We subsequently performed a proteogenomic analysis by integrating the transcriptome and proteome data to mine novel splice-junction events that may have contributed to stomach development. Using the strategy illustrated in (Fig. 4b), we generated an in-house fasta database that contained 16,602 hypothetical peptides with in silico trypsinization. Briefly, these tryptic peptides should have no missed trypsin cleavage sites, are not present in the mouse Ref-Seq proteome database and are 3-frame translated from two exons: one exon is a known exon in the mouse reference genome database, and the other exon is an unannotated exon with a novel junction site. By searching the mass spectra against the in-house fasta file, we identified 60 splice-junction peptides with ion scores greater than 30 at 1% peptide FDR (Fig. 4c and Supplementary Data 5) and manually verified their MS/MS spectra (Supplementary Data 6). Forty-nine peptides were derived from alternative splicing within the coding frame of a known gene. A cluster of novel gene products (CDKN1C, IGSF3, HDAC4, MAP4K4, CDC27, and MYH11) were significantly expressed in the early stages of stomach development (Fig. 4c), whereas other gene products (VSIG1, SORBS1, TLN1, ITGA7, KRT16, DTNA, TNXB, and TLN1) were expressed at relatively late stages (Fig. 4c). For example, VSIG1, three known isoforms (NM_030181, XM_006528630, and XM_006528631; Supplementary Data 4) were identified from RNA-Seq; moreover, two novel isoforms were identified from RNA-Seq and were validated by LC-MS/MS (Fig. 4c and Supplementary Data 5). The two main isoforms (NM_030181 and XM_006528630) were relatively abundant in the mouse stomach and were highly expressed at the later stages of development (Supplementary Fig. 4d), which is consistent with a previous study[48]. If validated, these previously undetected peptides and their associated protein isoforms will identify new players in stomach development and developmental biology.

**Correlation of stomach development and gastric cancer.** Tumorigenesis has been suggested to resemble a mis-regulated embryonic development process in a variety of ways[31], and this view is supported by experimental evidence from a limited number of genes/proteins[30]. The protein and mRNA atlas of the mouse stomach development dataset obtained in this study and a recently collected proteome dataset of diffuse-type gastric cancer obtained from 84 patients[50] enabled us to investigate the correlation of development and cancer at the proteome level.

Transcriptional regulators and signaling pathways are the most intensively studied subjects in organ development. Dysregulations of transcription factors and pathway regulators are the hallmarks

of cancer[58,59]. Our proteomic and transcriptomic approaches enabled the identification of 241 and 725 transcriptional regulators, respectively (Supplementary Fig. 5a, b and Supplementary Data 7). While the majority of the transcription factors were highly expressed in Ph1 (the early embryonic days) at the protein or RNA levels (Supplementary Fig. 5a, b), a few transcription factors were highly expressed in Ph2 and Ph3, suggesting that these transcription factors may contribute to organ maturation. Among the well-characterized transcriptional regulators related to organ development and cancer, most of them are more highly expressed at the early embryonic stages (Ph1) when cells are under rapid division and proliferation. For example, Barx1 is a mesenchymal gene and is restricted to the stomach mesenchyme during gut organogenesis;[5] CDX2 is a prognostic marker of gastric cancer[60], and its RNA was elevated in the early stages of stomach development; Pdx1 is an antrum epithelial specific gene that may be associated with intestinal metaplasia in the stomach[61,62]. In general, the expression of these transcription factors is well-correlated at the protein and RNA levels (Supplementary Fig. 5c, d). Nevertheless, one transcription factor (GATA4) exhibited opposite trends in the two data sets, thus providing an example that protein and RNA may not always be positively correlated.

When examining proteins or genes that are listed in the eight key signaling pathways known to be involved in stomach development, we identified a large number of proteins or genes significantly altered (ANOVA, FDR < 0.05) across the three phases at the protein or RNA levels (Fig. 5). Noticeably, a considerable number of these genes are also upregulated proteins (symbols in cyan) (Student's $t$-test, FDR < 0.05) or top mutated genes (italic symbols with underlines, mutation rate ≥ 5%) in diffuse gastric cancer (DGC). For example, WNT5A in the Wnt signaling pathway was significantly upregulated, while *Apc* and *Ctnnb1* were mutated in DGC. The observation on the alterations of these regulators in stomach development and DGC is a clear indication of the dysregulation of transcriptional regulators and signaling pathways in gastric cancer.

To further correlate stomach development with gastric cancer on the levels of signaling pathways, we listed all gene products that were detected in two studies and counted the number of significantly altered gene products in one of the eight signaling pathways known to be involved in development and cancer (Fig. 6a, Supplementary Data 8). Two-tailed hypergeometric statistical tests were carried out to determine whether there was a significant overlap of altered gene products on a specific pathway between stomach development and DGC. Importantly, all pathways with the exception of the hedgehog signaling pathway exhibited high correlations (-$\log_{10}P$-value > 3). The poor correlation for the hedgehog pathway was likely due to the small number of gene products detected in this pathway in the three data sets. These analyses provided evidence that stomach development and gastric tumorigenesis may be two related processes.

To obtain a global view of when the elevated proteins identified in DGC are expressed in stomach development, we selected 939 proteins that were differentially expressed between tumor and nearby tissues and then mapped the expression of their mouse homologs to the 15 timepoints during stomach development (Fig. 6b and Supplementary Data 8). A considerably higher percentage of proteins were identified in the early embryonic days (Ph1) than in the later timepoints (Fig. 6b). Altogether with the previously described findings, our results show that gastric cancer shared many characteristics with embryonic stomach development and suggest that the disease may result from the dysregulation of transcription factors and cell signaling.

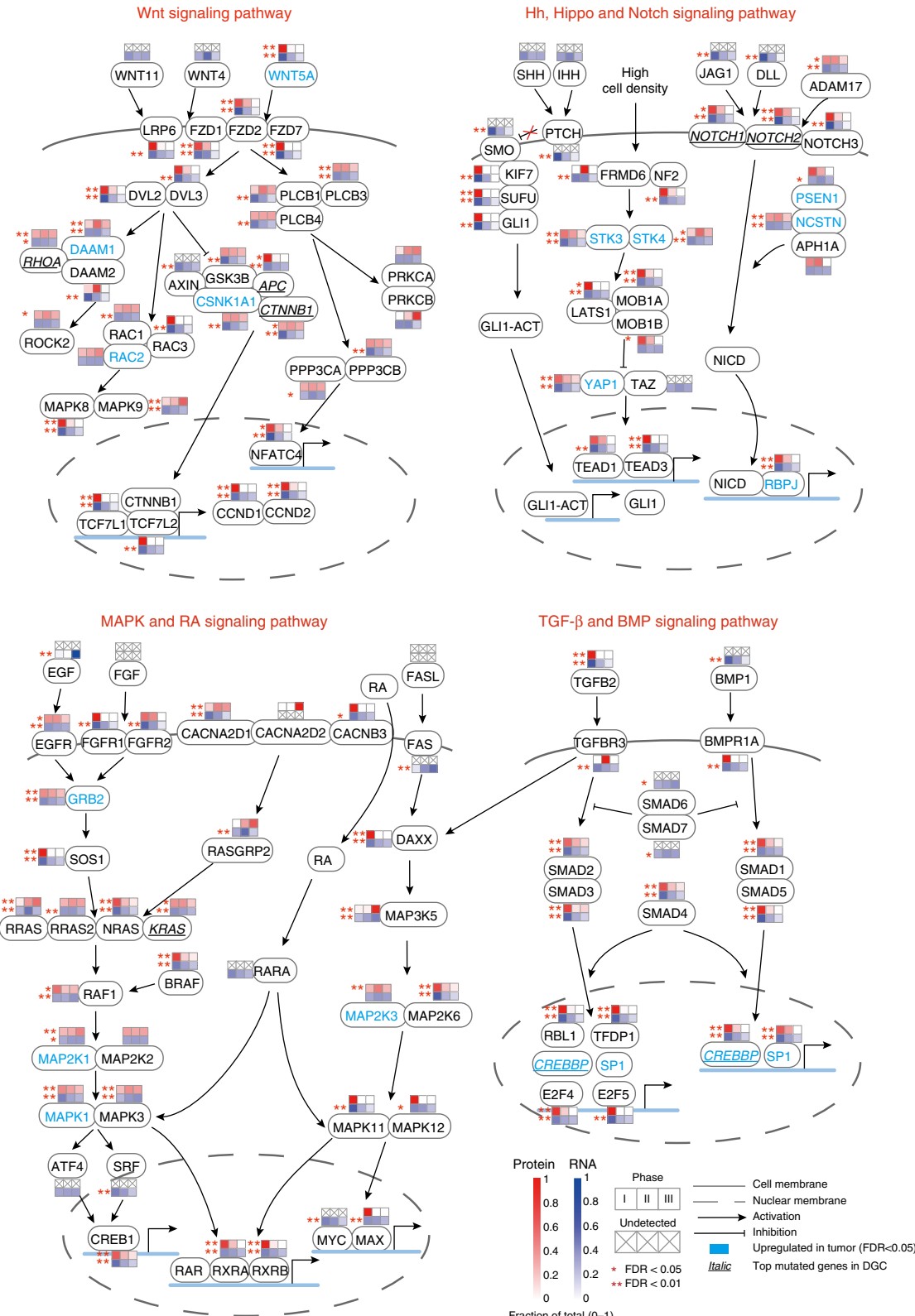

**Fig. 5** Key signaling pathways involved in stomach development and gastric cancer. Statistical analyses of proteomic and transcriptomic data to identify differentially expressed genes (small lower squares on each gene symbol) or proteins (small upper squares on each gene symbol) in stomach development. Proteins differentially expressed and genes frequently mutated in DGC were compared to those in stomach development. Cross marks represent genes or proteins that are not detected in the study; solid lines represent cell membranes; dashed lines represent nuclear membranes; lines with solid arrows illustrate activation of downstream genes; lines with hammer-heads illustrate inhibition of downstream genes; gene symbols in cyan are upregulated in tumor (FDR < 0.05); italic gene symbols with underlines are top mutated genes in DGC

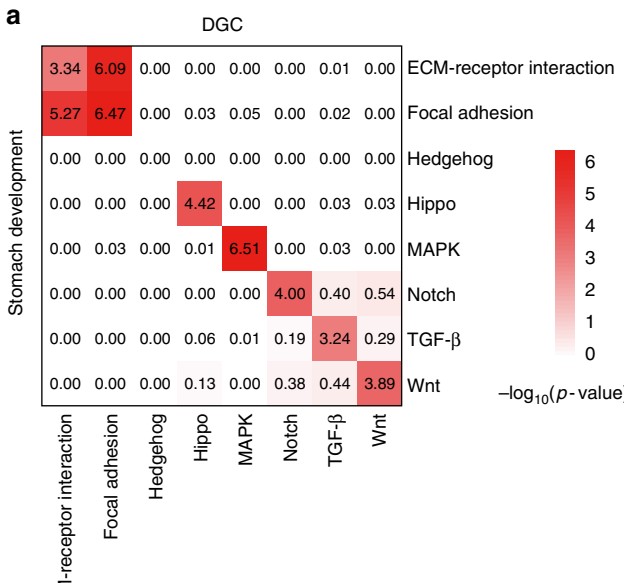

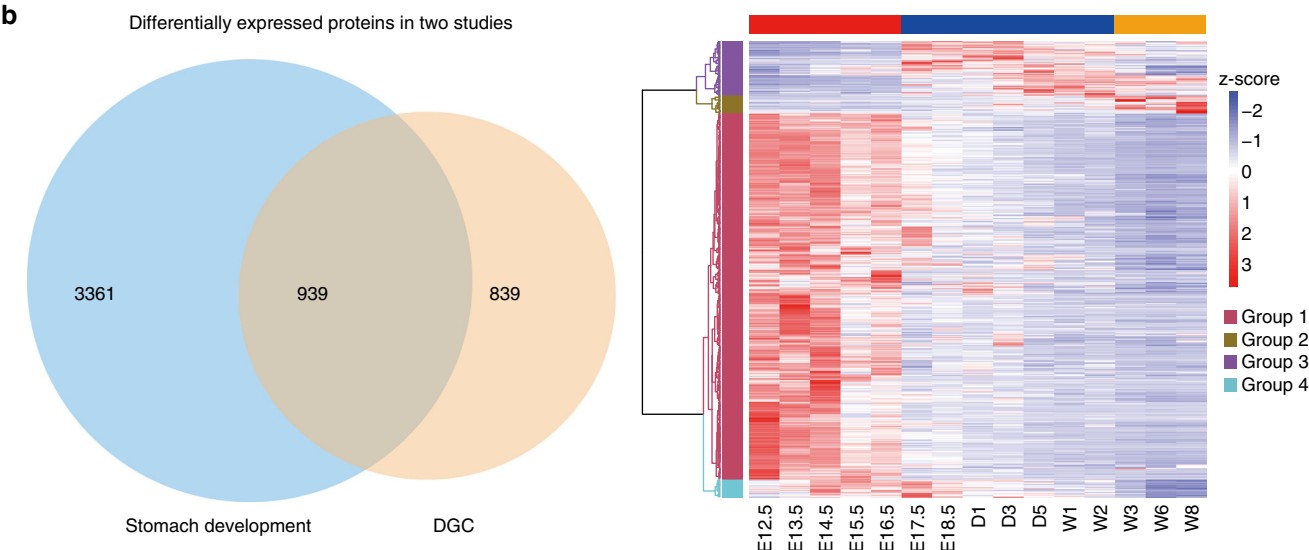

**Fig. 6** Comparative analyses of stomach development and gastric cancer. **a** Two-tailed hypergeometric tests were conducted to determine whether there is a significant overlap of altered genes on a specific pathway between stomach development and diffuse-type gastric cancer. Upper left: Correlations on eight signaling pathways between stomach development and DGC. Upper right: the number of significantly altered and overlapping genes on eight signaling pathways in stomach development and DGC. **b** Venn diagram of genes or proteins (ANOVA, FDR < 0.01) differentially expressed in stomach development and DGC (*t*test, FDR < 0.05, ratio of tumor over nearby tissues ≥ 2) (Lower left); Temporal expression patterns of 939 overlapping genes across 15 timepoints (Lower right). Right panel, horizontal color bars: red, Ph1; blue, Ph2; and orange, Ph3. Vertical color bars: deep pink, group 1; gold, group 2; purple, group 3; and cyan, group 4

## Discussion

The current study presents the first protein and mRNA panoramic overview of the developing mouse stomach with high proteome coverage and time resolution. This protein and mRNA atlas spanning the developmental period from embryo to adult provides a systems view for this process and permits data mining and knowledge generation as a public resource for other scientists.

Our preliminary bioinformatic analysis suggests that the mouse stomach development process from the embryo to adult may be classified into three phases based on protein and mRNA patterns. Both proteomic and transcriptomic data confirmed that the first phase, Ph1, is a period for RNA-associated transcription and cell division, and the last phase, Ph3, is a phase highlighted by increased metabolism and stomach maturation, while the second phase, Ph2, is somewhat associated with ECM-receptor interaction and focal adhesion. Correlating the proteomes or driver proteins/pathways of these three phases with the morphological, physiological, and functional states of the mouse stomach to understand development in greater molecular detail will be interesting. Coexpression analyses of proteins/mRNAs in different phases of the development revealed many understudied proteins/mRNAs with similar temporal expression patterns with well-studied proteins, which may shed light on the functions of these currently understudied or unknown proteins.

As the RNA-Seq technique is more mature than proteomics, mRNA data are generally considered to have deeper coverage than protein data. Furthermore, the correlation between protein

and RNA is known to be moderate. As proteins are the effector molecules, mRNA cannot be used as a replacement for protein. Current proteomic techniques and the experimental costs have started to permit the large-scale measurement of proteins. In this study, we found that a subset of genes or proteins was only identified in one of the data sets, which acknowledges the advantages and benefits of measuring both mRNA and protein in handling complex biological challenges.

We found that combining proteomic and transcriptomic data together enabled the detection and verification of alternatively spliced transcripts that may have fundamental impacts on stomach development. Our study not only generated a list of differentially expressed splicing isoforms but also detected peptides covering novel splicing junctions, providing stronger evidence for their existence at the protein level. Keep in mind that the number of these novel splicing events may be underestimated because some peptides are not readily detectable by LC-MS/MS.

One of the motivations of this study was to investigate whether there is experimental evidence for a connection between development and tumorigenesis. By comparing the proteomic profiling of stomach development with that of diffuse-type gastric cancer, we found that a noticeable number of pathway regulators, particularly transcription factors that are upregulated or frequently mutated in DGC, are also differentially expressed during stomach development. Moreover, we identified statistically significant sets of overlapping genes in seven signaling pathways between stomach development and gastric cancer, which suggests that DGC regains some features of stomach embryonic development but loses some basic functions of the adult stomach and that gastric cancer is likely a consequence of the dysregulation of master regulators and cell signaling pathways. It would be interesting to examine whether these features are also shared by the intestinal-type and mixed-type gastric cancers. However, this comparison should be treated with caution given that DGC develops largely from epithelial cells, while the whole stomach examined in this study is a mixture of multiple cell types. While a region-resolved stomach proteome could partially address this issue, this technique is hampered by technical challenges that the regions are discernable only around W2. Further investigations with a cell-type-resolved proteome could substantially improve the resolution and reduce the cellular heterogeneity, making the comparison of development and tumorigenesis more thorough.

In conclusion, the high-throughput multiomics data sets provide valuable assets that enable further data interrogation to obtain clues for a better understanding of stomach development and gastric cancer.

## Methods

**Animals and stomach collection**. C57BL/6 mice (8–10-weeks-old) were ordered from Beijing HFK Bioscience Co., LTD (Beijing China) and housed under a standard SPF (specific pathogen-free) laboratory environment. Whole stomachs were separated from mouse embryos during gestation or from the newborn mice after birth. Stomachs were collected at 15 timepoints covering the embryonic and postnatal stages: E12.5, E13.5, E14.5, E15.5, E16.5, E17.5, E18.5, D1, D3, D5, W1, W2, W3, W6, and W8. Each timepoint has three biological replicates.

The whole stomach tissues were washed twice with ice-cold phosphate-buffered saline (PBS) to remove blood and other contaminates, quick-frozen in liquid $N_2$ and stored at –80 °C for RNA or protein extractions. All animal experiments were approved by the animal care regulations of the Institutional Animal Care and Use Committee of the State Key Laboratory of Proteomics, Beijing Proteome Research Center, Beijing Institute of Radiation Medicine.

**Extraction and digestion of stomach proteome**. At least 1 mg samples of stomach tissues were cut off and lysed in a buffer that consisted of 8 M Urea, 100 mM Tris Hydrochloride, pH 8.0, protease and phosphatase inhibitors (Thermo Fisher Scientific, Rockford, IL, USA) for 10 min. The preliminary lysates were sonicated under a suitable condition. The lysates were centrifuged at 16,000 × $g$ for 10 min at 4 °C, and the supernatants were reserved as whole tissue extract (WTE). The protein concentration was determined by a Bradford protein assay. Approximately

100 μg of WTE were reduced with 10 mM dithiothreitol (DTT) at 56 °C and alkylated with 20 mM iodoacetamide (IAA) at room temperature in the dark. WTE was digested with sequencing grade trypsin that cleaves at the C-terminus of the Arg or Lys residues at 37 °C.

**Mass spectrometry analysis**. The tryptic peptides were vacuum-dried, redissolved in 10 mM of ammonium bicarbonate buffer (pH 10), and subjected to small-scale reversed-phase (sRP) chromatography with a homemade C18 column. The peptides were separated into nine fractions by stepwise increasing acetonitrile (ACN) from 6 to 35% under a basic condition (pH 10)[32,33]. These fractions were combined into six samples, vacuum-dried, and stored at –80 °C until subsequent use for liquid chromatography tandem mass spectrometry (LC-MS/MS) analysis.

MS samples were analyzed on a Q Exactive HF mass spectrometer (MS) (Thermo Fisher Scientific) interfaced with an Easy-nLC 1000 nanoflow LC system (Thermo Fisher Scientific). Briefly, the samples were redissolved in 30 μl of Solvent A (0.1% formic acid in water), and one fifth of the reconstituted samples were loaded onto a homemade reversed-phase C18 column (2 cm × 100μm; particle size, 3μm; pore size, 300 Å) and then separated by a 150μm × 12 cm silica microcolumn (homemade; particle size, 1.9μm; pore size, 120 Å) with a linear gradient of 5–35% Mobile Phase B (0.1% formic acid in acetonitrile) at a flow rate of 600 nl/min for 75 min. A data-dependent strategy was used by measuring MS1 in the Orbitrap at a resolution of 120,000 followed by tandem MS scans of the top 20 precursors using higher-energy collision dissociation with 27% of normalized collision energy and 18 s of dynamic exclusion time. Trypsin digests of 293T cells as quality control samples were routinely assayed to ensure good sensitivity and reproducibility.

**Protein identification and quantification**. MS Raw files were searched against the National Center for Biotechnology Information (NCBI) Ref-seq mouse proteome database (updated on 04/07/2013, 27,414 entries) in Proteome Discoverer workstation (version 2.0, Thermo Fisher Scientific, Rockford, IL, USA) implemented with Mascot search engine with percolator (Matrix Science, version 2.3.01). The following search parameters were used: (1) the mass tolerances were 20ppm for precursor ions and 50mmu for product ions; (2) up to two missed cleavages were allowed; (3) the minimal peptide length was seven amino acids; (4) cysteine carbamidomethylation was set as a fixed modification, and N-acetylation and oxidation of methionine were considered variable modifications; and (5) the charges of precursor ions were limited to + 2, + 3, + 4, + 5 and + 6. The data were also searched against a decoy database to estimate the peptide FDR using percolator validation based on the $q$-value. Although not evaluated directly, the protein FDR was minimized by choosing only proteins identified with one unique and two strict peptides. A label-free, intensity-based absolute quantification (iBAQ) approach[63] was used to calculate protein quantification based on the area under the curve (AUC) of precursor ions. The fraction of total (FOT) was used to represent the normalized abundance of a protein across experiments. The FOT was defined as a protein's iBAQ divided by the total iBAQ of all identified proteins in one experiment. The FOT was further multiplied by $10^5$ to obtain iFOT for the ease of representation. Missing values were substituted with zeros.

**RNA sequencing and data processing**. Total RNA was extracted from the liver tissues and treated with deoxyribonuclease I (DNase I). mRNA was isolated by Oligo Magnetic Beads and cut into small fragments that served as templates for cDNA synthesis. Once short cDNA fragments were purified, they were extended with single nucleotide adenines, ligated with suitable adapters, and amplified by PCR before they were sequenced. High-throughput RNA sequencing (RNA-Seq) experiments were carried out using the Illumina comprehensive next-generation sequencing (NGS) technique. Raw data were filtered and processed by the FastQC software (Version 0.11.5, Available online at website: http://www.bioinformatics. babraham.ac.uk/projects/fastqc). Low quality RNA-Seq reads were removed if they had a Phred quality score less than 20 or had less than 50 nucleotides. The filtered reads were mapped onto the mouse reference genome (GRCm38.p2.genome, released on 12/10/2013) using HISAT2 software (Version 2.1.0)[64]. Assembly and quantification of the transcripts were accomplished with StringTie software (Version 1.3.1) using the mouse genome annotation file as the reference (gencode.vM2. annotation, available at the GENCODE website)[65]. Fragments Per Kilobase of transcript per Million mapped reads (FPKM) was used for the measurements of the relative abundances of the transcripts. An effective expressed gene required an FPKM score greater than one, which resulted in 16,839 transcripts. The protein-mRNA correlation was calculated by Pearson's correlation coefficients using normalized scores ($z$-scores) across all timepoints.

To identify novel splicing junctions in this study, a customized python script was developed with the following strategies: (1) an unannotated exome was connected with a known exome; (2) these two exomes are in silico spliced and translated into peptides based on the 3-frame translation criterion; and (3) generated sequences are cut into tryptic peptides without missed trypsin cleavage sites and stop codons. The script generated 16,602 unique sequences from RNA-Seq, and a fasta file was subsequently created for mass spectrometry database searching.

**Bioinformatics and statistical analysis**. Principal component analysis (PCA) and unsupervised hierarchical clustering analysis were performed on 7,678 GPs in Dataset3 and grouped 15 timepoints into three phases. Gene Ontology (GO) term enrichment analysis was based on DAVID (the Database for Annotation, Visualization and Integrated Discovery) Bioinformatics Resources[66]. Differentially expressed genes in the three phases were identified by One-way ANOVA test with Benjamini-Hochberg adjustment. Gene coexpression analysis was conducted by Ward's hierarchical clustering analysis. Interexperiment correlations were calculated by Spearman's correlation coefficients. Two-tailed hypergeometric statistical tests were carried out to determine whether there is a significant overlap of altered genes on a specific pathway between stomach development and gastric cancer.

**Code availability**. The custom python script for mining splicing-junction peptides is available from the corresponding authors upon request.

**Reporting Summary**. Further information on research design is available in the Nature Research Reporting Summary linked to this article.

## Data availability

MS raw files and searching output data are deposited into proteomeXchange with the accession number PXD010702; RNA-Seq data are deposited into the NCBI Gene Expression Omnibus (GEO) database with the accession number GSE118083. A reporting summary for this Article is available as a Supplementary Information file. All other data supporting the findings of this study are available from the corresponding authors on reasonable request.

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

## Acknowledgements

This work was supported, in part, by grants from the National Program on Key Basic Research Project (973 Program, 2014CBA02000), the State Key Laboratory of Proteomics (SKLP-YA201401), the International Science & Technology Cooperation Program of China (2014DFA33160), the National Nature Science Foundation of China (31770892) and the National Key Research and Development Program of China (2017YFA0505102).

## Author contributions

X.L., C.Z., C.D., B.Z., J.X., Y.W., and J.Q. conceived the project; X.L., T.G., M.L., and L.S. performed the experiments; X.L., T.G., X.N., C.Z., J.L., and D.Z. analyzed the data; X.L., C.Z., Y.W., and J.Q. wrote the manuscript.

## Additional information

**Competing interests:** The authors declare no competing interests.

