## [Peer Review File · Nature Communications]

Reviewers' comments:

Reviewer #1 (Remarks to the Author):

This manuscript describes a high resolution analysis of the developing stomach at both the protein and RNA levels and compared the expression patterns to publically available proteome data on diffuse gastric cancer. The manuscript provides an excellent resource for the research community on proteins involved with different stomach developmental stages. A key observation is the description of three principle developmental stages, as defined by both the proteome and RNA data. This is strong and convincing data. Novel splice variants which may play a role in stomach development were also identified.

The comparison with the DGC proteome identified that the major signaling pathways were active in both the developing stomach and DGC. It is difficult to know whether this observation just reflects the proliferative activity of (any) cancer, or whether it really illuminates the nature of DGC. This could be answered if a similar comparison was able to be conducted with the intestinal type GC proteome. In the absence of that comparison, the correlations between the developing stomach and DGC lack weight, and can at best only suggest that DGC has its origins in the disruption of key developmental pathways. This is an idea worth confirming though! The methods are well described, but I cannot comment on the accuracy of the statistical methods- although I note they aren't a critical part of the manuscript.

Reviewer #2 (Remarks to the Author):

Review of NCOMMS-18-18383

This paper describes a comprehensive characterization of the proteome and transcriptome of the developing mouse stomach.

Pluses – I did not see anything in either how the experiments were carried out, nor in the interpretation of the results, that seemed incorrect. Fairly standard approaches, informatics tools, and statistical evaluations were employed. It is not that common for proteomics studies to enhance their depth by making use of transcriptomic data, so it was good to see the authors doing that. Their strategy for doing so seemed reasonable and straightforward, and is consistent with other published studies of that sort.

The study did generate an impressive amount of proteomic and transcriptomic data.

Minuses – The english, while generally clear, suffers a bit from the usual non-native english speaker grammatical errors. A thorough copy-editing by a native english speaker would be helpful.

I personally do not find this sort of study to be terribly interesting: it is a data generation exercise, and succeeded in generating a lot of data. It is difficult to gain much insight into any biological questions however, by simply generating a lot of data. However, I did look up the publication criteria for Nature Communications, which are:

- The data is technically sound
- The paper provides strong evidence for its conclusions

- The results are novel
- The manuscript is important to scientists in the specific field

And I would say that this paper does meet all of these criteria. Regarding the last one, I imagine there is a community of people out there that studies stomachs, and having access to a comprehensive data set of the proteins expressed in stomachs, and knowing how those expression patterns change during development, would seem to be something that community would want. The analysis of the data appeared to be quite thorough, I do not have anything to suggest that would add to it.

Here are a few (not a complete list) grammatical issues:

Abstract line 26 insert "of" before "stomach"

Line 31 I don't really know what a "longitudinal scale" is, and the sentence doesn't seem to say very much and/or is unclear.

Introduction line 59 "knowledge of stomach.." not "on stomach"; and "knowledgeis" not "knowledge..are"; line 60 "focus hampers" not "focus hampered"; line 61 I don't know what "in context of the global spectrum" is supposed to mean, it doesn't mean anything to me. What spectrum? Line 66 "to identify" not "for identifying"; line 75 "do they hold" not "do they held"; but "exhibit" would be a better word choice. line 76 delete "to each other"; line 80 delete "hence" ; line 85 cancer is not a process, it is a disease. "Development" is indeed a process.

Results line 155 "predominantly" is not spelled correctly (given as "predominately"); line 171 "stomach starts" not "started". Line 186 "Proteasome is... and are multifunctional enzymes" sentence is not right... what are multifunctional enzymes? Line 242 "expression" not "expressions" - this mistake repeated multiple places in the ms. Line 260 and 261 need to add word "exon" after "known" and "unannotated". Line 276 insert "a" before "limited" ; line 285 insert "the" before "majority" ; line 286 the "on the protein or RNA levels" is incorrectly written but needs more than a 1 word fix to repair. Line 294 statement that proteins and RNA may be negatively correlated is at odds with other statements in the paper stating the opposite trend. Lines 314 and 315 same cancer and development issue, they are not both processes. Line 316 replace "an" with "a"; line 324 "result" not "be resulted"

Discussion line 327 "presents" not "presented" ; line 328 "sufficient" is unclear. Sufficient for what? ; line 333 delete "an" ; line 341 "detail" not "details"; line 345 delete "getting" ; line 347 "executioners" is not the right word.... Executioners are people that kill other people. Try "agents"?; line 363 "attention" not "attentions"

Line 476 "Inter-experiment" not "Inter-experiments" ; "Ward" should be capitalized as a proper name.

In Figure 1b panel II "Sample" is misspelled.

Reviewer #3 (Remarks to the Author):

This is essentially a resource paper, as the authors have undertaken a large-scale, first-of-its-kind analysis of the proteome and transcriptome of the stomach from early development to adult. Mostly, the data are fine, and all the tables will, as mentioned, be a key resource for the field. The analysis of such a massive amount of data must, of course, be limited to what could be done with it, though they make some effort to validate expression of known markers, to analyze RNA-protein expression differences, to find new alternative splice and alternate transcript expression levels,

and then to care to a subset of gastric cancer. There are no real novel conclusions per se about stomach biology or development, though I don't think that is necessary.

The one major caveat which really has to be addressed is the unsophisticated and unthorough discussion of how the data are on WHOLE stomach at multiple times, which means that all 3 germ layers are contributing in varying ways at various times. This is particularly a deficiency when comparing to tumor databases, which are largely composed of a single cell type (the epithelial tumor itself).

Specifically, this is a problem as follows:

It is not clear how well the authors understand anatomy/histoanatomy of the stomach:

1) most important, they make no attempt to sort out which transcripts/proteins are expressed in which part of the stomach: forestomach (lined by squamous epithelium in rodents), corpus/body (containing enzyme and acid secreting cells, antrum/pyloris (simpler, mucus-secreting glandular cells).

2) nor do they attempt to comment on which cells within which tissues are being analyzed. Not only are all anatomic regions lumped together, endoderm-derived epithelium and mesoderm-derived muscle and ectoderm-derived enteric nervous system cells are all lumped together

3) let alone the various epithelial cell types that are arising at the time points they study, starting with a simple columnar, undifferentiated population to the complex, multicellular adult populations.

Given that the tumor data will be largely from epithelial cells, the comparison between the whole stomach and the tumors has limitations (eg. In Fig. S5). For example, Barx1 is a mesenchymal gene in stomach development, whereas Gata4 is epithelial, and Cdx2 is not expressed in the stomach at all, yet these are all mixed together in Fig. S5. CDX2 is a marker of gastric tumors because gastric tumors arise out of metaplastic events that involve transdifferentiation or dedifferentiation to a more primitive epithelium that may have intestinal characteristics. Thus, Fig. 5 and this discussion should be in the contexts of the caveats.

Examples of particularly confusing statements:

-GIF is a glycoprotein expressed by parietal cells (this is not true in mice, wherein GIF is in chief cells)

-The paper the authors cite about VSIG1 emphasizes its role in early stomach development, yet it is increased much more strongly in later stages in the authors' data. The authors of the cited study talk about differential expression of isoforms of Vsig1. Do the authors confirm those results? There are alternative splicing peptides identified in Fig. 4c. How do those jibe with the previous study?

-Barx1 see above

- Pdx1 is an antrum epithelial specific gene but this isn't contextualized

-This statement makes no sense at all: "The glandular stomach differentiates into smooth muscle, the muscle layer thickens at E14.5, and finally different types of cells become mature." glands become muscle"?

-"The mouse stomach is composed of three main parts: forestomach for food storage, corpus for food digestion, and antrum that secretes endocrine." The antrum does more than secrete hormones ("secretes endocrine" is a mistake)

Other points:

Standard nomenclature for mouse proteins is ALL CAPS, no italics, same as humans, not first letter cap.

What does it mean that "P2" has the least number of up-regulated proteins? That is confusing. Upregulated relative to what? I assume what the authors are trying to say is that there are few

proteins that are more highly expressed at that stage than the other stages, implying it is a transitional stage without any specific proteins that define it but, rather, lower abundance of the preceding and following stages.

The analysis of Mediator and the proteasome is not well introduced. Why is this important? The authors state: "The deep proteome coverage provides an opportunity to observe the dynamics of key components in a protein complex." But then they do no additional analysis other than list the proteins that are expressed and change. Also, it is a little tough to compare the RNA-Seq and protein patterns in separate figures. Is there a way to represent the relative correlation between the two so that lines 230-237 are a little clearer?

More explanation of Fig. 4a in the text is warranted. How was the correlation done? Relative levels of expression overall across all timepoints or was it a binary call at each timepoint, etc.? The data in fig. 3c look remarkably similar actually, more so than I expected so what makes for the poor correlation in 4a?

What is a "dark" protein in line 344?

Lots of typos and non-standard grammar, eg:

"smample" in Fig. 1

"Introduction The mammalian stomach is a part of the muscular sac and characteristic curves of the proximal digestive tract."

"is the fast-growing phase when stomach changed most in shape and mass."

The "D" and "W" in "D1, W1" etc. should be defined in the text as "day" and "week".

"As the main organ of the digestive system, there are many digestive enzymes that" – dangling modifier.

"To survive in the acidic environment, stomach secretes mucin proteins .." - missing definite article (the stomach).

"Proteasome is another multi-subunit protein complex and are multifunctional enzymes." Unclear wording.

"There are other gastric proteins (Anxa10, Cldn18, Ctse, Ghrl, Gif, Tff1, and Vsig1) that are highly expressed in stomach" Aren't gastric and stomach the same thing?

"The first big branch of dendrogram can be further cut into two trees to separate the " missing "the"

"Majority of genes that are differentially expressed in the first phase are enriched in pathways such as spliceosome" missing "the"

"While majority of the transcription factors were highly expressed in P1 (early embryonic days) either on the" missing "the"

"To obtain an global view" "an" should be "a"

Reviewers' comments:

Reviewer #1 (Remarks to the Author):

This manuscript describes a high resolution analysis of the developing stomach at both the protein and RNA levels and compared the expression patterns to publically available proteome data on diffuse gastric cancer. The manuscript provides an excellent resource for the research community on proteins involved with different stomach developmental stages. A key observation is the description of three principle developmental stages, as defined by both the proteome and RNA data. This is strong and convincing data. Novel splice variants which may play a role in stomach development were also identified.

The comparison with the DGC proteome identified that the major signaling pathways were active in both the developing stomach and DGC. It is difficult to know whether this observation just reflects the proliferative activity of (any) cancer, or whether it really illuminates the nature of DGC. This could be answered if a similar comparison was able to be conducted with the intestinal type GC proteome. In the absence of that comparison, the correlations between the developing stomach and DGC lack weight, and can at best only suggest that DGC has its origins in the disruption of key developmental pathways. This is an idea worth confirming though! The methods are well described, but I cannot comment on the accuracy of the statistical methods-although I note they aren't a critical part of the manuscript.

Response: We thank this reviewer for the positive comments. We agree that, with the DGC data alone, the observation could just reflect the proliferative activity of any cancer. We are currently in the process of analyzing intestinal type GC (IGC) proteome. To answer the reviewer's question, we performed a similar analysis using 50 IGC proteomes to compare with the developing stomach (**Figure CL1**). Our results showed an overlap of proteins in 8 major pathways in developing stomach with those upregulated IGC (**Figure CL1-A**). These data suggest that the disruption of key development pathways is likely characteristics of both DGC and IGC. In the main text, we changed the description for this observation as gastric cancer (GC) instead of DGC. We also added this analysis as a possible scenario in the discussion. The IGC data are presented here for this reviewer only and are not included in the current manuscript since the IGC dataset will be analyzed in its entirety in a separate manuscript.

Figure CL1 Comparative analyses of stomach development and intestinal type gastric cancer

(A) Two-tailed hypergeometric tests were conducted to address if there is a significant overlap of altered genes on a specific pathway between stomach development and 50 cases of IGC. Upper left: Correlations on eight signaling pathways between stomach proteome and IGC. Upper right: the number of significantly altered and overlapping genes on eight signaling pathways in developing stomach and IGC. (B) Venn diagram of genes or proteins that differentially expressed in stomach development (ANOVA, FDR<0.01) and IGC (t test, FDR<0.05, ratio of tumor over nearby tissues ≥ 2) (Lower left); Temporal expression patterns of 903 overlapping gene products across 15 timepoints (Lower right).

Reviewer #2 (Remarks to the Author):

This paper describes a comprehensive characterization of the proteome and transcriptome of the developing mouse stomach.

Pluses – I did not see anything in either how the experiments were carried out, nor in the interpretation of the results, that seemed incorrect. Fairly standard approaches, informatics tools, and statistical evaluations were employed. It is not that common for

proteomics studies to enhance their depth by making use of transcriptomic data, so it was good to see the authors doing that. Their strategy for doing so seemed reasonable and straightforward, and is consistent with other published studies of that sort. The study did generate an impressive amount of proteomic and transcriptomic data.

Minuses – The english, while generally clear, suffers a bit from the usual non-native english speaker grammatical errors. A thorough copy-editing by a native english speaker would be helpful.

I personally do not find this sort of study to be terribly interesting: it is a data generation exercise, and succeeded in generating a lot of data. It is difficult to gain much insight into any biological questions however, by simply generating a lot of data. However, I did look up the publication criteria for Nature Communications, which are:

- The data is technically sound*
- The paper provides strong evidence for its conclusions*
- The results are novel*
- The manuscript is important to scientists in the specific field*

And I would say that this paper does meet all of these criteria. Regarding the last one, I imagine there is a community of people out there that studies stomachs, and having access to a comprehensive data set of the proteins expressed in stomachs, and knowing how those expression patterns change during development, would seem to be something that community would want. The analysis of the data appeared to be quite thorough; I do not have anything to suggest that would add to it.

Here are a few (not a complete list) grammatical issues:

Abstract line 26 insert “of” before “stomach”.

Line 31 I don’t really know what a “longitudinal scale” is, and the sentence doesn’t seem to say very much and/or is unclear.

Introduction line 59 “knowledge of stomach..” not “on stomach”; and “knowledge ...is” not “knowledge..are”; line 60 “focus hampers” not “focus hampered”; line 61 I don’t know what “in context of the global spectrum” is supposed to mean, it doesn’t mean anything to me. What spectrum? Line 66 “to identify” not “for identifying”; line 75 “do they hold” not “do they held”; but “exhibit” would be a better word choice. line 76 delete “to each other”; line 80 delete “hence” ; line 85 cancer is not a process, it is a disease. “Development” is indeed a process.

Results line 155 “predominantly” is not spelled correctly (given as “predominately”); line 171 “stomach starts” not “started”. Line 186 “Proteasome is... and are multifunctional enzymes” sentence is not right... what are multifunctional enzymes?

Line 242 “expression” not “expressions” - this mistake repeated multiple places in the ms. Line 260 and 261 need to add word “exon” after “known” and “unannotated”. Line 276 insert “a” before “limited” ; line 285 insert “the” before “majority” ; line 286 the “on the protein or RNA levels” is incorrectly written but needs more than a 1 word fix to repair. Line 294 statement that proteins and RNA may be negatively correlated is at odds with other statements in the paper stating the opposite trend. Lines 314 and 315 same cancer and development issue, they are not both processes. Line 316 replace “an” with “a”; line 324 “result” not “be resulted”.

Discussion line 327 “presents” not “presented”; line 328 “sufficient” is unclear. Sufficient for what?; line 333 delete “an” ; line 341 “detail” not “details”; line 345 delete “getting” ; line 347 “executioners” is not the right word.... Executioners are people that kill other people. Try “agents”?; line 363 “attention” not “attentions”.

Line 476 “Inter-experiment” not “Inter-experiments”; “Ward” should be capitalized as a proper name.

In Figure 1b panel II “Sample” is misspelled.

Response: We have corrected the language errors the reviewer pointed out and highlighted the major changes in yellow. The entire manuscript was also edited by professional manuscript services.

Reviewer #3 (Remarks to the Author):

This is essentially a resource paper, as the authors have undertaken a large-scale, first-of-its-kind analysis of the proteome and transcriptome of the stomach from early development to adult. Mostly, the data are fine, and all the tables will, as mentioned, be a key resource for the field. The analysis of such a massive amount of data must, of course, be limited to what could be done with it, though they make some effort to validate expression of known markers, to analyze RNA-protein expression differences, to find new alternative splice and alternate transcript expression levels, and then to care to a subset of gastric cancer. There are no real novel conclusions per se about stomach biology or development, though I don't think that is necessary.

The one major caveat which really has to be addressed is the unsophisticated and unthorough discussion of how the data are on WHOLE stomach at multiple times, which means that all 3 germ layers are contributing in varying ways at various times. This is particularly a deficiency when comparing to tumor databases, which are largely composed of a single cell type (the epithelial tumor itself).

Specifically, this is a problem as follows:

It is not clear how well the authors understand anatomy/histoanatomy of the stomach:

1) Most important, they make no attempt to sort out which transcripts/proteins are expressed in which part of the stomach: forestomach (lined by squamous epithelium in rodents), corpus/body (containing enzyme and acid secreting cells), antrum/pyloris

(*simpler, mucus-secreting glandular cells*).

Response: We agree that a region-resolved stomach proteome is more valuable to understand the functions of the different parts of the stomach; however, it is not technically feasible to do so for majority of the time points since the size of the stomach is too small. In a previously published study, we indeed attempted to divide the stomach into three parts (forestomach, corpus, and antrum) and profiled the region-specific proteome in mature (~20-week-old) mice [Figure CL2, based on data from this study and the article "Effect of high fat diet on proteome in mouse stomach", Chinese Journal of Biotechnology, Oct. 25,

2018, 34(10), <http://journals.im.ac.cn/cjbcn>, DOI: 10.13345/j.cjb.180085, in Chinese].

As shown in the Venn diagram below, while majority of proteins (56.7%) are expressed in all three regions, each region has distinct expression pattern and contains region-specific proteins, likely reflecting in part the contribution from different cell types that carry out specific functions.

Figure CL2 Region-specific proteomes from the three parts of stomach

(A) Venn diagram of number of proteins in forestomach, corpus and antrum. (B) Region-specific protein expression patterns in forestomach, corpus and antrum. Unsupervised hierarchical clustering was performed for 3 independent replicates.

2) *Nor do they attempt to comment on which cells within which tissues are being analyzed. Not only are all anatomic regions lumped together, endoderm-derived epithelium and mesoderm-derived muscle and ectoderm-derived enteric nervous system cells are all lumped together.*

Response: We recognize that the lack of cell-type resolved proteomes is a limitation of this study. Unlike single cell RNA-Seq, single cell proteome is extremely challenging and can provide only limited information from small number of proteins.

There is also a lack of adequate cell surface markers that allow for sorting of different cell types. Nonetheless, we made attempts to discuss some known cell-type specific proteins with the limited information in Figure 2d.

3) Let alone the various epithelial cell types that are arising at the time points they study, starting with a simple columnar, undifferentiated population to the complex, multicellular adult populations. Given that the tumor data will be largely from epithelial cells, the comparison between the whole stomach and the tumors has limitations (e.g. In Fig. S5). For example, Barx1 is a mesenchymal gene in stomach development, whereas Gata4 is epithelial, and Cdx2 is not expressed in the stomach at all, yet these are all mixed together in Fig. S5. CDX2 is a marker of gastric tumors because gastric tumors arise out of metaplastic events that involve transdifferentiation or dedifferentiation to a more primitive epithelium that may have intestinal characteristics. Thus, Fig. S5 and this discussion should be in the contexts of the caveats.

Response: We are aware of the reviewer's concerns. The limitations and caveats are fully discussed in the manuscript. Please see the highlighted sections in the *Discussion*.

Examples of particularly confusing statements:

-GIF is a glycoprotein expressed by parietal cells (this is not true in mice, wherein GIF is in chief cells)

Response: Corrected.

-The paper the authors cite about VSIG1 emphasizes its role in early stomach development, yet it is increased much more strongly in later stages in the authors' data. The authors of the cited study talk about differential expression of isoforms of Vsig1. Do the authors confirm those results? There are alternative splicing peptides identified in Fig. 4c. How do those jibe with the previous study?

Response: We identified 3 known isoforms (NM_030181, XM_006528630, XM_006528631) (Figure CL3-A) and 2 novel isoforms from RNA-Seq data and validated the novel isoforms at peptide level by LC-MS/MS. The NM_030181 and XM_006528630 are highly expressed in the mouse stomach. The protein expression starts to be detectable at E15.5, and is stably detected at E17.5 (Figure CL3-B). The two novel isoforms have never been reported before, and they are also highly expressed from E18.5 and beyond.

The study by Oidovsambuu et al reported the identification of 3 Vsig1 transcripts, 2 (Vsig1A and Vsig1B) were expression in stomach, 1 (Vsig1C) was expressed in the testis. The time course of the expression of individual transcript was not reported, but the total RNA blot showed that the expression was increased postnatal (Figure CL3-C), consistent with our RNA and protein expression data. The authors then used an anti-VSIG1 immunofluorescence to show the restricted expression in the glandular

epithelium of the stomach at E12.5. They also showed by Western blotting the expression of transgenic EGFP-Vsig from E15.5 to P20, and the endogenous VSIG1 from E18.5 to P60 (Figure CL3-D). Overall, our data agree with this study that stable detection of VSIG1 protein expression starts at late embryonic days, but differ in that the expression level is increased postnatal. Additionally, the immunostaining appears to be more sensitive than MS in detecting VSIG1 protein at early embryonic phase. The authors also generated an X-linked Vsig inactivation mouse model, which showed that VSIG1 is required for the proper differentiation of glandular gastric epithelia (which we quote in the main text).

Figure CL3 Expression of VSIG1 in different experiments

(A, B) The expression patterns of the *Vsig1* transcripts (A) and the VSIG1 protein (B) (our data). (C, D) The expression of *Vsig1* at the RNA level (C) and the protein level (D) (reproduced from: Oidovsambuu O, Nyamsuren G, Liu S, Göring W, Engel W, et al. (2011) Adhesion Protein VSIG1 Is Required for the Proper Differentiation of Glandular Gastric Epithelia. PLOS ONE 6(10): e25908. <https://doi.org/10.1371/journal.pone.0025908>).

-*Barx1* see above

Response: Barx1 has been discussed in the manuscript as suggested.

-*Pdx1* is an antrum epithelial specific gene but this isn't contextualized

Response: PDX1 has been discussed in the manuscript as suggested.

-This statement makes no sense at all: "The glandular stomach differentiates into

smooth muscle, the muscle layer thickens at E14.5, and finally different types of cells become mature." glands become muscle?

Response: Modified as “Cells that originate from the corpus and antrum start to differentiate into smooth muscle, the muscle layer thickens at E14.5, and different types of cells ultimately become mature”.

"The mouse stomach is composed of three main parts: forestomach for food storage, corpus for food digestion, and antrum that secretes endocrine." The antrum does more than secrete hormones ("secretes endocrine" is a mistake)

Response: Modified as “The mouse stomach is composed of three main parts: the forestomach for food storage, the corpus for food digestion, and the antrum that secretes mucus and certain hormones”.

Other points:

Standard nomenclature for mouse proteins is ALL CAPS, no italics, same as humans, not first letter cap.

Response: Changed as suggested.

What does it mean that “P2” has the least number of up-regulated proteins? That is confusing. Upregulated relative to what? I assume what the authors are trying to say is that there are few proteins that are more highly expressed at that stage than the other stages, implying it is a transitional stage without any specific proteins that define it but, rather, lower abundance of the preceding and following stages.

Response: The term “up-regulated proteins” is inaccurate and is now replaced with “differentially expressed proteins”. They refer to proteins whose expression is statistically higher in one phase than the other two phases (ANOVA, FDR < 0.01).

The analysis of Mediator and the proteasome is not well introduced. Why is this important? The authors state: “The deep proteome coverage provides an opportunity to observe the dynamics of key components in a protein complex.” But then they do no additional analysis other than list the proteins that are expressed and change. Also, it is a little tough to compare the RNA-Seq and protein patterns in separate figures. Is there a way to represent the relative correlation between the two so that lines 230-237 are a little clearer?

Response: The background of the mediator and the proteasome are incorporated into the manuscript, and detailed discussion of the dynamic change of these large protein machinery are added (See highlighted section in - *A Combined Analysis of mRNA and Protein Reveals Protein-RNA Correlation and Novel Splice-junction Events* and the modified Fig. S4b, c).

More explanation of Fig. 4a in the text is warranted. How was the correlation done? Relative levels of expression overall across all timepoints or was it a binary call at

each timepoint, etc.? The data in fig. 3c look remarkably similar actually, more so than I expected so what makes for the poor correlation in 4a?

Response: More explanation of Fig. 4a is provided in the figure legend. Protein-mRNA correlation was calculated by Pearson's correlation coefficients using normalized scores (z-scores) for each gene product across all time points. (More details in section - *RNA Sequencing and Data Processing*). Regarding the inconsistency of Fig. 3c and Fig. 4a, enrichment analysis (Fig. 3c, now changed to Fig. S3d in the revised version) and correlation analysis (Fig. 4a) are different bioinformatic algorithms. Enrichment analysis aims to identify a group of genes or proteins that are over-represented in a large set of background genes/proteins, whereas the correlation analysis evaluates the strength of a relationship of a gene or protein in different groups. A list of genes and their products from both RNA-Seq and MS may be enriched in a specific pathway but the correlations of them may be weak if they changed different across time points. However, some pathways (e.g. spliceosome) that are both enriched in protein and RNA also present a good correlation between the two datasets.

What is a "dark" protein in line 344?

Response: "dark" proteins refer to functionally unknown proteins. To avoid ambiguity, we reworded it ("dark proteins") as understudied or unknown proteins.

Lots of typos and non-standard grammar

"smapple" in Fig. 1

"Introduction-The mammalian stomach is a part of the muscular sac and characteristic curves of the proximal digestive tract."

"is the fast-growing phase when stomach changed most in shape and mass."

The "D" and "W" in "D1, W1" etc. should be defined in the text as "day" and "week".

"As the main organ of the digestive system, there are many digestive enzymes that" – dangling modifier.

"To survive in the acidic environment, stomach secretes mucin proteins .." - missing definite article (the stomach).

"Proteasome is another multi-subunit protein complex and are multifunctional enzymes." Unclear wording.

"There are other gastric proteins (Anxa10, Cldn18, Ctse, Ghrl, Gif, Tff1, and Vsig1) that are highly expressed in stomach" Aren't gastric and stomach the same thing?

"The first big branch of dendrogram can be further cut into two trees to separate the" missing "the".

"Majority of genes that are differentially expressed in the first phase are enriched in pathways such as spliceosome" missing "the".

"While majority of the transcription factors were highly expressed in P1 (early embryonic days) either on the" missing "the".

"To obtain an global view" "an" should be "a".

Response: We have corrected all typos and grammar errors (changes are highlighted in yellow) and the entire manuscript has been edited by the Springer Nature Author Services.

REVIEWERS' COMMENTS:

Reviewer #3 (Remarks to the Author):

The authors have done a good job of addressing my concerns.